# Anti-Alopecia Activity of Alkaloids Group from Noni Fruit against Dihydrotestosterone-Induced Male Rabbits and Its Molecular Mechanism: In Vivo and In Silico Studies

**DOI:** 10.3390/ph15121557

**Published:** 2022-12-14

**Authors:** Laila Susanti, Resmi Mustarichie, Eli Halimah, Dikdik Kurnia, Andi Setiawan, Yustinus Maladan

**Affiliations:** 1Department of Pharmaceutical Analysis and Medicinal Chemistry, Faculty of Pharmacy, Universitas Padjadjaran, Jatinangor 45363, Indonesia; 2Department of Pharmacy, Faculty of Mathematics and Natural Science, Universitas Tulang Bawang, Bandar Lampung 35128, Indonesia; 3Department of Pharmacology and Clinical Pharmacy, Faculty of Pharmacy, Universitas Padjadjaran, Jatinangor 45363, Indonesia; 4Department of Chemistry, Faculty of Mathematics and Natural Science, Universitas Padjadjaran, Jatinangor 45363, Indonesia; 5Department of Chemistry, Faculty of Mathematics and Natural Science, Universitas Lampung, Bandar Lampung 35141, Indonesia; 6Eijkman Research Center for Molecular Biology, The National Research and Innovation Agency, Bogor 16915, Indonesia

**Keywords:** androgenic alopecia (AA), dihydrotestosterone (DHT), Noni fruit (*Morinda citrifolia* L.), anagen/telogen ratio, molecular docking dynamics simulation

## Abstract

Androgenic alopecia (AA) is a condition that most commonly affects adult men and is caused by an increase in the hormone dihydrotestosterone (DHT) in the hair follicles. Anti-alopecia drugs should be discovered for hair follicles to enter the anagen growth phase. Therefore, this study evaluated the hair growth-promoting activity of Noni fruit’s water, ethyl acetate, *n*-hexane fractions, and sub-fractions from the active fraction in the alopecia male white rabbit model. The Matias method was modified by inducing rabbits using DHT for 17 days, followed by topical application of Noni fruit solution for 21 days. Meanwhile, hair growth was evaluated by histological observation of the follicular density and the anagen/telogen (A/T) ratio in skin tissue. In the first stage, five groups of male white rabbits were studied to obtain the active fraction; DHT+Minoxidil as standard, DHT+vehicle (NaCMC 1%), DHT+FW, DHT+FEA, and DHT+FH. The FEA as the active fraction was followed by open-column chromatography separation (DCM:Methanol) with a gradient of 10% to produce sub-fractions. In the second stage, the six main sub-fraction groups of male rabbits studied were DHT+FEA-1 to DHT+FEA-6. The follicular density of groups FEA-3 was 78.00 ± 1.52 compared with 31.55 ± 1.64 and 80.12 ± 1.02 in the Vehicle and Minoxidil groups. Additionally, group FEA-3 showed large numbers of anagen follicles with an A/T ratio of 1.64/1 compared to the vehicle group of 1/1.50 and 1.39/1 for Minoxidil control. Group FEA-3 was identified by LC-MS/MS-QTOF, followed by molecular docking to the androgen receptor (PDB: 4K7A), causing alopecia. The results showed that three alkaloid compounds with skeleton piperazine and piperidine, namely (compounds **2** (−4.99 Kcal/mol)**, 3** (−4.60 Kcal/mol), and **4** (−4.57 Kcal/mol)) had a binding affinity similar to Minoxidil, with also has alkaloid skeleton piperidine–pyrimidine (−4.83 Kcal/mol). The dynamic behavior showed the stability of all androgen receptor compounds with good RMSD, SMSF, and SASA values after being studied with 100 ns molecular dynamics (MD) simulations. This study produced a common thread in discovering a class of alkaloid compounds as inhibitors of androgen receptors that cause alopecia.

## 1. Introduction

Alopecia is a scalp disorder with partial or complete hair loss. This is caused by hair falling out more than the amount of growth. It is not a severe disorder but can cause mental health problems and decrease quality of life [1]. Androgenic alopecia (AA) is genetically linked to an imbalance in the amount of dihydrotestosterone (DHT) hormone binding to androgen receptors in hair follicles. As a result, it causes the miniaturization of hair follicles to stop growth [2,3]. This miniaturization causes the anagen (growth phase) follicle to become inactive, shorten, and for the outer root sheath to shrink. The change becomes the start of the catagen phase (transition phase) and continues with the telogen phase (resting phase), with the hair follicles in the stationary phase.

Treatment for AA which has been approved by the Food and Drug Administration (FDA) is topical Minoxidil, which belongs to vasodilator drugs with the mechanism of action being dilating blood vessels. Following its application, blood flow becomes smooth, and hair follicles become active to allow regrowth. However, Minoxidil has long-term side effects, such as skin irritation, itching, and redness. It is dangerous for pregnant and lactating women.

The profile of side effects from alopecia treatment has motivated research on alternative medicine from herbal plants with low toxicity and side effects. They can also provide nutritional support for hair growth, inhibiting the hormone DHT and the enzyme 5-α-reductase, or as aromatherapy in increasing blood circulation in the scalp [4].

In this study, the selected herbal plants were Noni fruit (*Morinda citrifolia* L.), a part of the Rubiaceae family (Figure 1). This plant grows well in tropical climates, such as the Pacific Islands, Southeast Asia, and Hawaii [5]. Noni fruit has been used as a traditional folk medicine and is commonly consumed as an herb in the form of juice [6]. More recently, Noni fruit has been used as a functional food supplement, such as Noni fruit flour as a dietary supplement [7] and fermented Noni fruit as a treatment for acute liver injury due to alcohol consumption [8]. Several scientific studies report Noni fruit as anti-viral [9], anti-bacterial [10], anti-diabetic [11], anti-inflammatory [12], and anti-hypertensive [13]. While other studies on Noni fruit are popular in systemic diseases, its effects on skin treatment are yet to be widely reported, especially regarding hair growth. Noni fruit is ethnobotanically used as a mask to treat hair loss and dandruff [14]. Existing related research only covers the extract of Noni fruit as anti-dandruff to inhibit 14 out of 18 isolates of fungi that cause dandruff (77.78%) [15]. The use of this fruit to treat dry dandruff 1 × 1 day is proven to reduce itching, dandruff, and hair loss compared to 1 × 2 days [16]. Several studies have reported anti-alopecia activity using the shaving method (in vivo), reporting bioactive compounds, flavonoids, polyphenols [17,18], palmitic acid [18], and anthraquinones [19]. In vitro studies on dermal papilla cells have also reported the bioactive compound properties of alkaloids [20,21]. Furthermore, the content of Noni fruit compounds has also been reported to contain flavonoids [22,23], anthraquinones [24], and alkaloids [25].

In the present study, in vivo studies used male white rabbits as a preclinical test induced by DHT. DHT, a more potent androgen hormone than testosterone, can lead to hair miniaturization, decreased hair density, and continued hair loss. DHT 10^−6^ mol/L has been able to inhibit the growth of hair follicles, entering the catagen phase faster for 17 days [26]. In the previous literature, loss of hair from the dorsal of mice was visible after a 21-day induction with testosterone [27,28,29]. Quantitative observations are required to establish the diagnosis by identifying the follicles in the skin biopsy specimen through histology analysis. A method of analyzing data was carried out to observe follicular density (number of follicles/mm^2^) and the follicle anagen and telogen ratio (A/T ratio). The method of anti-alopecia activity on male white rabbits using dihydrotestosterone (DHT) hormone induction and skin biopsies is still not widely used, while other studies have reported anti-alopecia activity (in vivo) using the shaving method and analyzed data only from the length of hair growth. The search for active compounds from Noni fruit as anti-alopecia is essential, and assay-guided separation has needed further experiments to the sub-fraction stage.

The mechanism of anti-alopecia activity from in vivo results is complemented by the in silico molecular mechanism approach. It has emerged as a crucial technique in understanding and deciphering macromolecular structure-to-function relationships [30,31]. Molecular docking, molecular dynamics, and ADME-Tox modeling represent the three most widely used components of computational modeling and have been crucial in supporting information in vitro and in vivo studies [32,33]. Parameters such as binding affinity, H-bond interaction, surface H-bond, RMSD (Root Mean Square Deviation), RMSF (Root Mean Square Fluctuation), and SASA (Solvent Accessible Surface) have been observed to estimate the therapeutic efficacy of a candidate drug [34]. Other in silico approaches to predict absorption, distribution, metabolism, excretion, and toxicity (ADME-Tox) are needed to clarify the pharmacokinetics and pharmacodynamics of compounds [35]. We conducted these two studies (in vivo and in silico) to emphasize the importance of the molecular approach from the results of laboratory experiments to support drug discovery. In addition, the molecular mechanism approach of Noni fruit has not been reported to the androgen receptor with the PDB ID code 4K7A.

As described above, this study aimed to evaluate the anti-alopecia activity of Noni fruit compounds by quantitative observation of anagen and telogen hair follicles on male rabbits induced by the DHT hormone. Another objective is to identify and describe the structural relationship of compounds to androgen receptors using a molecular mechanism method. The data obtained from this study are significant as initial information for developing anti-alopecia drugs from Noni fruit.

## 2. Results

### 2.1. Standardization of Noni Fruit Extract

Botanists conducted plant determination at the Botanical Laboratory, Faculty of Mathematics and Natural Sciences, Universitas Lampung. According to Cronquist (1981) and APG II clarification systems, the determination of plant tests refers to *Morinda citrifolia* L. A total of 4.2 kg of ripe Noni fruit was extracted with 96% ethanol to produce a viscous extract yield of 14.65%. The crude extract (615.3 g) was carried out with a liquid partitioning with *n*-hexane-H_2_O and ethyl acetate-H_2_O, resulting in the yields of 3.5%, 37.21%, and 49.23% fractions of *n*-hexane (FH), ethyl acetate (FEA), and water (FW), successively. The viscous extract of Noni fruits was analyzed for specific and non-specific parameters according to the Indonesian Herbal Pharmacopoeia Standards [36] (Table 1). 

Phytochemical screening of ethanolic extract of Noni fruit on alkaloids, flavonoids, phytosterol, anthraquinones, saponins, and tannins showed positive results. Meanwhile, FW and FEA screening showed positive results for alkaloids, saponins, and tannins. The FH was positive for phytosterols and anthraquinones (Table 2).

### 2.2. Anti-Alopecia Activity of Fraction and Sub-Fraction: In Vivo Studies

Histologically transverse sections of the affected skin area were observed using an ocular micrometer after administration (Figure 2 and Figure 3). Quantitative observation in skin tissue 1 × 1 mm^2^ produced data on hair follicular density (the total of hair follicles, anagen and telogen) and A/T ratio (anagen and telogen ratios) (Table 3). 

In the first stage, for FW and FEA fractions (Groups III and IV), the follicular density was 69.12 ± 1.35 and 76.78 ± 0.84 compared with 80.12 ± 1.02 for the Minoxidil control (Group II). Groups III and IV also showed many anagen follicles with an A/T ratio of 1.35/1 and 1.84/1, respectively. On the other hand, the FH (Group V) gave an effect of follicular density and A/T ratio data comparable to the vehicle group (Group I) (Table 3). Statistically, the FW and FEA fractions had the same activity, but the FEA fraction had a higher follicular density and anagen follicles. The observation also supports this result that the baldness area of FEA (Group IV) after administration was lower than FW and FH (Groups III and V), which means that the FEA sample succeeded in promoting hair growth (Table 4). Therefore, this study focused on the more active fraction. We specified the FEA for further experiments by open-column chromatography (OPC) to generate sub-fractions, which were further tested for anti-alopecia activity by the same method.

In the second stage, the sub-fraction of FEA-3 (Group VIII) resulted in hair follicular density and an A/T ratio of 78.00 ± 1.52 and (1.64/1) (Table 3). This group activated telogen to anagen follicles towards optimal hair growth, which was indicated by the number of anagen follicles by a decrease in the average area of baldness after the treatment samples from 3.10 cm^2^ to 1.72 cm^2^ (Table 4). Groups VI, VII, and IX produced more anagen follicles but had low follicular density, and Groups X–XI produced more telogen follicles and low follicular density. One-way ANOVA confirmed this result. The value of *p* < 0.05 showed a significant difference between groups, followed by the Least Significant Difference (LSD) test at the fraction stage (Groups III–IV), and the sub-fraction stage (Group VIII) was not significantly different from Group II (Minoxidil). Therefore, we specified FEA-3 (Group VIII) as the active sub-fraction group, to be further investigated by LC-MS/MS analysis followed by in silico analysis (molecular docking, molecular dynamics, and ADME-Tox properties).

### 2.3. Area of Baldness in the Alopecia Rabbit Model

Figure 4 and Figure 5 summarize the areas of baldness after DHT induction and solution sample administration. On the results of DHT induction (s.c.) after 17 days, groups of fraction and sub-fraction showed changes in hair loss to alopecia with an average index of 3.18 ± 0.16 and 3.15 ± 0.12, respectively (Table 4). Therefore, DHT induction has caused alopecia to develop in the surrounding posterior area of more than 2 cm^2^ [18]. After alopecia occurred, Groups I–V were treated with 0.2 mL of the topical solution fraction group twice daily for 21 days, and the baldness area of the fractioned sample decreased from 3.18 ± 0.16 to 2.07 ± 0.17. The changes in the average area of baldness after treatment of sample solution in the sub-fraction group also showed a decrease of 3.15 ± 0.12 to 1.97 ± 0.12. Thus, the sub-fraction group from the OPC method of FEA triggered better hair growth. The results of the one-way ANOVA after the treatment of a sample group stated a significance value of *p* < 0.05, followed by the LSD test showing that the area of baldness of all groups was not significantly different, except for Group I.

### 2.4. Anti-Alopecia Activity: In Silico Studies (Molecular Docking, Molecular Dynamics, ADME-Tox)

In vivo test statistics showed that the sub-fraction of FEA-3 (group VIII) from the FEA fraction of Noni fruit had anti-alopecia activity against DHT-induced test animals. LC-MS/MS was used to investigate the profile of active compounds in the FEA-3 sub-fraction, which gave six prominent peaks. We have identified the LC-MS/MS compound structure depicted in Table 5 and Table 6 and Figure 6 and Figure 7. Therefore, a molecular mechanism approach was conducted from the compounds FEA-3 sub-fraction against androgen receptors that cause alopecia. A selected receptor was 4k7A, an androgen receptor that develops male sexual phenotypes. The method was validated by removing natural ligands (DHT and Minoxidil) at the androgen 4K7A receptor with grid box x = −2.592, y = 0.864, z = −6.729 producing an RMSD value of 1.732 Å, which is smaller than the data bank of 2.44 Å.

The LC-MS/MS energy positive ionization chromatogram of the sub-fraction FEA-3 showed a high abundance at retention times of 9.94 and 11.38 min with *m*/*z* observed at 343.2945 and 371.3281, respectively (Appendix A). Abundant composition appeared at retention times of 12.86 (*m*/*z* 399.3608), 14.33 (*m*/*z* 427.3687), and 15.21 min (*m*/*z* 669.3326), while the low abundance composition appeared at retention times of 4.56 min (*m*/*z* 169.0765) (Table 5). Tracking with the built-in library database instrument revealed a fascinating fact—a group of chemical structures with an alkaloid skeleton was found in the FEA 3 sub-fraction, such as indole, piperazine, piperidine, oxazolidine, and pyrimidine (Table 6). Interestingly, by comparing the skeletal structure of the compound Minoxidil (standard), we found another fact—Minoxidil also contains an alkaloid skeleton of piperidine–pyrimidine.

The molecular docking of Compounds **1–6** and Minoxidil as a standard binding affinity (Kcal/mol), hydrogen bond, hydrogen bonds’ distance, and nearest amino acid residues are shown in Table 7. Compounds **2, 3,** and **4** produced the lowest binding affinity with visualization of amino acid interactions and H-bond surfaces compared to Minoxidil as the standard (Figure 8).

The molecular docking revealed that all Compounds **1–6** could bind to cofactor binding sites on protein receptors with a negative binding affinity. Meanwhile, Compound **1** did not have H-bonds; it has the nearest amino acid residues in common with other groups, namely, Glu793, Gln858, Leu862, and Lys861. We focused on Compounds **2**, **3**, and **4**, which produced binding affinities (−4.99, −4.60, and −4.57 Kcal/mol) slightly different from Minoxidil as a standard (−4.83 Kcal/mol) to be studied further by visualization of the amino acid interaction and H-bond surface presented in Figure 8.

Molecular dynamics simulation was carried out for 100 ns (100.000 ps) to predict the dynamic behavior between the androgen receptor (4K7A) and the three best compounds (Compounds **2**, **3**, and **4**) through RMSD, RMSF, and SASA parameters. The stability assessment of the complex was analyzed based on the RMSD criteria described in Figure 9. The interactions of all compounds with the androgen receptor were relatively stable during the 100 ns simulation with complex fluctuations below 0.20 nm resulting in an average RMSD value of 0.135 nm. The interaction of the androgen receptor with Minoxidil has the same RMSD average value as Compound **3,** which is 0.150 nm. Meanwhile, Compound **3** increased RMSD 0.198 nm at 70 ns but slowly decreased to match Minoxidil at 90 ns. Interestingly, the interaction between the androgen receptor and Compound **2** was relatively stable from 0 ns to 100 ns with a RMSD value of 0.110 nm, while other compounds reached stability at 90 ns. Generally, the pattern between androgen receptors with all compounds did not differ significantly, indicating that the androgen receptor structure is stable during interactions with Compounds **2**, **3**, and **4**.

An amino acid residue flexibility assessment is influenced by the type of ligand interacting with the receptor represented by the RMSF graph (Figure 9). Fluctuation represents the flexibility of amino acid residues. Compound **3** resulted in a higher fluctuation of amino acid residues than other compounds recorded in Ser851. There were fluctuations in some residues, such as Gln693, His729, and Asp819, of about 0.12 nm. The Ser851 residue showed a very high peak intensity of Compound **3** with a fluctuation value of about 0.4 nm. Then the fluctuation increased slightly in Minoxidil by around 0.13 nm with the recorded amino acid residue Ser778. Generally, the RMSF residue in the androgen receptor region showed that all complexes had similar oscillations.

SASA analysis was performed for each complex to calculate the surface area of the androgen receptor accessible to the solvent. The small surface area indicates the stability of the compound during the simulation because the accessibility of water molecules to the receptor–compound complex is reduced. The mean scores of surface area between the complexes of androgen receptor–Minoxidil, androgen receptor–Compound **2,** androgen receptor–Compound **3**, and androgen receptor–Compound **4** were 122.71, 120.21, 121.49, and 120.64 nm^2^, respectively (Figure 10). Solvent access in the androgen receptor–Compound **2** is the smallest (120.21 nm^2^), which indicates it is more stable than other complexes. However, the score surface area of all complexes is similar to Minoxidil (standard).

The pharmacokinetic parameter of human intestinal absorption (HIA) of all compounds is >80%, which means that these compounds can be well-absorbed (range 70–100%) in the human intestine [37]. A skin permeability (SP) score with log Kp < 2.5 indicates the good potential activity of the compound when administered topically [38]. Compound **2** and Minoxidil have better permeability to the skin than Compounds **3** and **4**. All compounds with a blood–brain barrier (BBB) with a value of >0.3 were predicted to be able to penetrate across BBB [39]. The drug candidate’s plasma protein binding (% PPB) level influences the drug’s action, properties, and efficacy. Compounds **2** and Minoxidil have 43.48% and 58.38% PPB, lower than Compounds **3** and **4**. This result explains that Compounds **3** and **4**, with PBB scores of 93.21% and 95.13%, can diffuse through the plasma membrane and interact with proteins [40]. However, the toxicity test results of all compounds were not carcinogenic. The standard ADME-Tox properties of Compounds **1–6** and Minoxidil are presented in Table 8.

## 3. Discussion

The solvent’s polarity determines the yield amount; in Noni fruit extract, the FW and FEA yielded higher than FH, which indicated that the Noni fruit compound content was polar to semi-polar; these results are relevant to previous studies [41,42,43]. According to the Indonesian Herbal Pharmacopoeia Standards, the standardization of extracts provided pharmacologically measurable efficacy and ensured user safety. Phytochemical screening revealed differences in compound content based on solvent polarity. Alkaloids, flavonoids, saponins, and tannins appeared in the FW and FEA, while phytosterols and terpenoids appeared in the non-polar *n*-hexane fraction. Noni fruit extract with ethanol solvent has provided an extract that complies with the standards. Previous studies have shown the presence of flavonoids and alkaloids in the FEA but not containing saponins [44]. In this study, saponins appeared in the FW and FEA, and this was due to using ripe fruit.

Dihydrotestosterone (DHT), a more potent androgen than testosterone, results in the miniaturization of the hair follicle and change in the cyclic phase, which leads to androgenic alopecia [26]. Pathogenically, alopecia occurs because androgen hormones enter the follicular papillae through capillaries. They are metabolized to DHT by the 5α-reductase enzyme and bind to androgen receptors abundant in hair follicles. Strong binding between DHT receptors causes gene expression to change, the anagen phase (hair growth phase) to become shorter, and the telogen (resting phase) to become more extended, which causes the miniaturization of hair follicles. This causes terminal hair to become smooth and unpigmented. Androgens profoundly affect the growth of the human scalp and body hair but lead to androgenetic alopecia (AA) loss in males [45].

Table 4 and Figure 5 reveal that after DHT induction, the area of baldness in the fraction and sub-fraction stage enlarged by an average of 3.18 ± 0.16 and 3.15 ± 0.12 cm^2^, respectively. However, the area of baldness became smaller and varied after being given different topical solutions in each stage. At the sub-fraction stage, there was a significant decrease in the area of baldness (3.15 ± 0.12 cm^2^ to 1.97 ± 0.12 cm^2^) compared to the fractional stage after treatment, especially in Group VIII (3.10 ± 0.06 cm^2^ to 1.72 ± 0.04 cm^2^), which indicated that the topical solution was more effective for hair growth. The results of skin morphological analysis confirmed this fact.

We summarized the skin morphological results of the fraction and sub-fraction stages in Figure 3. Good hair growth is characterized not only by a large number of follicles, but also by the type of anagen (growth phase) that must dominate over telogen (resting phase), and their appearances differ considerably. Anagen follicles are characterized by fully developed follicles within and outside the root sheath, and the suprabulbar zone begins to differentiate [46]. Telogen follicles are characterized by central hair canal wrinkling, necrosis, more destroyed follicles, and follicular shrinkage [46] (Figure 4). The abundant number of telogen follicles refers to the miniaturization of follicles towards alopecia. However, firstly, we performed the histological observations of the fractional stage presented in Figure 3 (Groups I, II, III, IV, and V, which statistically resulted in Group IV (FEA)) and found it did not differ significantly from Group II (Minoxidil). Then, we performed histological observations on the OPC samples from the FEA group presented in Figure 3 (VI, VII, VIII, IX, X, and XI), which resulted in the best Group VIII (FEA-3). Group VIII (FEA-3) triggers the formation of hair follicles and makes more anagen follicles. Meanwhile, another group had few hair follicles and telogen predominated, implying that the DHT induction process caused follicle miniaturization and continued alopecia. Previous studies reported an increase in density and anagen follicles after being given petroleum ether and ethanolic extract *Cuscuta reflexa* Roxb solutions [29], and *A. capillus-veneris* 2%, as well as petroleum ether extract of *Phyllanthus niruri* 2% [28,47]. In addition, the FEA of *Sansevieria trifasciata* P. also increased follicle anagen [48].

The results of the molecular docking group FEA-3 sub-fraction revealed that Compounds **2, 3,** and **4** are potentially inhibitors of androgen receptors and have the potential for anti-alopecia treatments (Table 7). The visualization of the ligand–receptor surface was observed to determine the preference for a proton donor or acceptor in the H-bond (Figure 7). The H-bond between the receptor–Minoxidil is purple, similar to Compounds **2** and **3**. It occurs intermolecularly due to the proton donor process between His789-(**H---N**)–Minoxidil, His789–(**H---O**)–Compound **2,** and Arg854–(**H---O**)–Compound **3**. Meanwhile, it occurs intermolecularly due to the proton acceptor process (green) between Trp796–(**N---H**)–Compound **2**, Glu793–(**N---H**)–Compound **2**, Trp796–(**N---H**)–Compound **4**, and Glu793–(**N---H**)–Compound **4**. Interestingly, this type of H-bond is the proton donor process between the receptor–ligand and involves similar amino acids, namely, His789 and Arg854. In contrast, the proton acceptor process between the receptor–ligand involves similar amino acids, namely, Trp796 and Glu793.

H-bonds play an essential role in protein–ligand binding, and they are also reported to promote ligand-binding affinity by displacing protein-bound water molecules into a bulk solvent [49,50]. The difference in the type of H-bonds is a factor that causes the binding energy of Compound **2** to be lower than Minoxidil, where the type of H-bonds on receptor–Compound **2** is –(**H---O**)- which is more polar than receptor–Minoxidil –(**H---N**)-. The Pauling scale states the difference in electronegativity of the –(**H---O**)- bond is 1.4 times greater than the -(**H---N**)- bond of 0.9, and polarity increases with an increasing difference in electronegativity [51]. The shorter H-bonds’ distance on the receptor–Compound **2** were 1.97, 1.86, and 2.16 Å, giving the H-bonds more potency in maintaining the stability of the ligand–receptor conformation when compared to 2.69 Å of the receptor–Minoxidil. Hydrophobic interactions on amino acid residues also determine the stability of ligands with androgen receptors. This is because hydrophobic interactions are more likely to agglomerate in the spherical structure of proteins [52]. The similar five amino acid residues interact hydrophobically, namely, Gln858-Tyr857-Leu862-Lys861-Leu797 in Minoxidil and Compounds **2–4,** indicating these amino acids play an essential role in the stability between the ligand and the androgen receptor. Hydrophobic interactions between inhibitor compounds and androgen receptors have also been reported in Leu862, Tyr857 [53] and Lys861, Trp796, and Leu797 [54].

The molecular dynamic simulation confirmed that all complexes had good stability, with RMSD values below 0.2 nm. The RMSD pattern results also correlated with SASA, which revealed that Compound **2** had better stability during simulation than other complexes. The amino acid fluctuations of the four receptor complex systems calculated by RMSF had similar oscillations, with residues Gln693, His729, Ser 778, Asp819, and Ser851 showing higher fluctuations, which is responsible for the loop region in the protein structure. Prediction of pharmacokinetics and the toxicity of chemical compounds can reduce failures to develop new drug discoveries. The parameter of HIA is considered a critical component that influences bioavailability, and serious efforts have been conducted to obtain an accurate prediction [37]. Generally, Minoxidil and Compounds **2–4** have ADME-Tox properties that are safe for the skin because anti-alopecia drugs are generally applied topically to the skin.

The molecular docking test results confirmed that Compounds **2, 3,** and **4** are named 1-[4-(2-Hydroxyethyl)-1-piperazinyl]-3-[(2-isopropyl-5-methylcyclohexyl)oxy]-2-propanol, 11-[(1-Hydroxy-2,2,6,6-tetramethyl-4-piperidinyl)(methyl)amino]undecanoic acid, and 2-Methyl-2-propanyl [(3S,4S,6S)-4-hydroxy-6-tridecyl-3-piperidinyl]carbamate, which acts as an anti-alopecia against androgen receptors. The LC-MS/MS tracking found new facts that these compounds have an alkaloid skeleton, such as piperazine and piperidine (Table 6). Interestingly, Compounds **1** (indole), **5** (oxazolidine), and **6** (pyrimidine) also have a basic alkaloid skeleton. Meanwhile, these compounds have a high binding affinity value compared to Minoxidil.

These results are highly relevant to the standard compound, Minoxidil (3-hydroxy-2-imino-6-piperidine-1-ylpyrimidine-4-amine), C_9_H_15_N_5_O, a piperidine–pyrimidine derivative, also an alkaloid group. Previous studies reported that the main compound of *Crinum asiaticum*, Norgalanthamine, C_16_H_19_NO_3_, an alkaloid, can increase hair growth through the proliferation of dermal papillae [20,21]. Based on in silico tests and observations of the skeletal structure of Compounds **2**, **3**, **4,** and Minoxidil, the alkaloid group plays a role in anti-alopecia. In a cyclic system, they are characterized by amine (NH_2_, -RNH-, and -NR_2_) or amide groups (-CO-NR_2_). The structural activity of the alkaloid group, which acts as an anti-alopecia, is due to the presence of a lone pair of electrons from the N-atom, which binds to other adjacent functional groups, and causes hydrogen, hydrophobic and electrostatic interactions.

## 4. Materials and Methods

### 4.1. Materials

*Morinda citrifolia* L. was collected from Lampung Province in Indonesia, and plant determination by Botanists from the Botanical Laboratory at Universitas Lampung. The ethanol 96%, NaCMC 1%, ethyl acetate, *n*-hexane, and distilled water purchased from Merck were used for extraction and fractionation. Furthermore, a dihydrotestosterone-2,3,4-^13^C_3_ (DHT) solution for induction was purchased from Sigma-Aldrich Pte Ltd, Aqua Pro Injection for the solution of DHT from Otsuka, and Minoxidil as a standard from PT. Surya Medica Laboratories. Silica gel 60 (0.063–0.200 mm), dichloromethane (DCM), methanol (MetOH), and Thin-Layer Chromatography (TLC) silica gel 60 F_254_ purchased from Merck were used for open-column chromatography (OPC) and identification of the eluate. NIKON Eclipse C*i*-L Plus was used to observe skin tissue (Tokyo, Japan). LC-MS/MS QTOF identified the compounds of the best sub-fraction from the active fraction with the specification LC system ACQUITY UPLC^®^H-Class System, LC column ACQUITY UPLC^®^HSS C18 (1.8 µm 2.1 × 100 mm) (Waters, Milford, MA, USA), and mass spectrometer Xevo G2-S QTOF (two-generation quadrupole time-of-flight) (Waters, USA). Meanwhile, the operating system Windows 10 Home Single Language 64-bit (10.0, Build 19041), RAM 8 GB, 3500U (8 CPUs), processor AMD Ryzen 5, and hard disc drive 500 GB were used to run the molecular docking process. Tensor TWS-1686525-GRO, Intel Xeon Gold 5115 Processor (double processor), RAM 192 GB, 512 GB 2.5 SATA III Internal Solid State Drive (SSD), 8 TB Memory, NVIDIA RTX 3080, and OS Centos 7 were used to run the molecular dynamics process.

### 4.2. Preparation and Standardization of Extract

Ripe Noni fruit was extracted using the maceration method with 96% ethanol at room temperature. The viscous extract was standardized for specific parameters of organoleptic profile, yield percentage, and phytochemical screening. Non-specific parameters include moisture content and total ash content. Dragendorf, Mayer, and Wagner tests conducted phytochemical screening of alkaloids. Meanwhile, Shinoda, Liebermann–Burchard, Brontager, Foam, and Ferric Chloride tests were used for flavonoids, phytosterols, anthraquinones, saponins, and tannins. The identifications were accomplished using standard methods [55,56]. The ethanolic extract was partitioned between *n*-hexane–water and ethyl acetate–water. Finally, the water (FW), ethyl acetate (FEA), and *n*-hexane (FH) fractions were re-screened for phytochemicals with those test series.

### 4.3. Open-Column Chromatography (OPC)

Preliminary TLC analysis on the active fraction of FEA was carried out. The plates were sprayed with H_2_SO_4_ 10% reagent in ethanol solvent to find the appropriate solvent system. The optimal solvent system of dichloromethane (DCM) and methanol (MetOH) was determined by a gradient of 10% with silica gel 60 (0.063–0.200 mm) as the stationary phase, resulting in 11 sub-fractions. The sub-fractions having a similar separation profile on TLC were combined based on observations on UV lamps at 254 nm, 356 nm, and sprayed with H_2_SO_4_ 10%. The mass of the combined sub-fractions was calculated as yield. The six main sub-fractions were obtained from the combined sub-fractions with similar separation profiles on TLC. The FEA-1 (combined sub-fractions 1–3; 32% *w*/*w* yield), FEA-2 (combined sub-fraction 4–5; 20% *w*/*w* yield), FEA-3 (sub-fraction 6; 15% *w*/*w* yield), FEA-4 (combined sub-fraction 7–8; 13% *w*/*w* yield), FEA-5 (sub-fraction 9; 11% *w*/*w* yield), and FEA-6 (combined sub-fraction 10–11; 9% *w*/*w* yield).

### 4.4. Animal

The Institutional Ethical Committee of Universitas Padjadjaran (No. 611/UN6.KEP/EC/2021) approves the protocol for all animal experiments. The test animals used 18 male white rabbits (5–6 months, 1.5–2 kg) purchased from Veterinary Centers Lampung, Indonesia. They were acclimatized for one week, provided feed and water ad libitum, and did not show more than 10% weight loss.

### 4.5. Alopecia Rabbit Model

The method reported by Matias was followed by minor alterations [26,27,28,29]. All white male rabbits were induced with DHT in sterile Aqua Pro Injection subcutaneously (s.c.) on the posterior back skin with a dose of 0.01 mg/0.1 mL once for 17 days. Alopecia symptoms were characterized by hair growth failure and widespread baldness after 17 days. The index of alopecia given to the DHT-induced area during observation of baldness are 0 (no hair loss occurs), 1 (diffuse thinning along the interscapular area), 2 (alopecia occurs in an area of 1 cm^2^), 3 (alopecia develops in the surrounding posterior area of 2 cm^2^), and 4 (alopecia is greater than or equal to 4 cm^2^) [27]. In addition, the alopecia index was measured quantitatively by measuring the area of baldness (cm^2^) and calculating the average. A representative photograph of rabbits shows the extent of alopecia after 17 days of DHT induction, described in Appendix A.

### 4.6. Anti-Alopecia Activity of Fraction and Sub-Fraction: In Vivo Studies

Preliminary studies have been carried out on an ethanolic extract of Noni fruit with concentrations of 5%, 15%, and 25% resulting in higher follicular density and the amount of anagen follicles at a concentration of 25%. A 0.2 mL solution was administered topically on the posterior back skin twice daily for 21 days. In the fractionation stage, the alopecia rabbit model was divided into five groups and treated as follows: (I) DHT(s.c.)+vehicle; (NaCMC 1%); (II) DHT(s.c.)+Minoxidil as standard; (III) DHT(s.c.)+FW 25%; (IV) DHT(s.c.)+FEA 25%; and (V) DHT(s.c.)+FH 25% resulted in higher follicle density and the number of anagen follicles belonging to the FEA. This fraction was separated by the OPC method resulting in six main groups.

In the sub-fraction stage, the alopecia rabbit models was divided into six groups and treated as follows: (VI) DHT(s.c.)+FEA-1 25%; (VII) DHT(s.c.)+FEA-2 25%; (VIII) DHT(s.c.)+FEA-3 25%; (IX) DHT(s.c)+FEA-4 25%; (X) DHT(s.c.)+FEA-5 25%; (XI) DHT(s.c.)+FEA-6 25%; from OPC separation of the FEA (active fraction) were also administered topically 0.2 mL on the posterior back skin twice daily for 21 days.

### 4.7. Histology Treatment

The rabbits were sacrificed by intramuscular anesthetic injection after 21 days of each activity stage (fraction and sub-fraction). A skin biopsy was performed on the affected area and stored in a BNF 10% solution for paraffin sectioning. Transverse skin tissue sections were prepared, dehydrated with alcohol, embedded, and blocked in paraffin. The microtomies used were adjusted to a thickness of 3–6 µm, and the slides were stained with hematoxylin–eosin. Histological preparations were examined under an ocular micrometer with 10× and 40× magnification (NIKON, Tokyo, Japan). The changes observed included degeneration and cyclical necrosis of hair follicles. The number in an area was recorded and reported as hair follicle density (number of follicles/mm^2^), and the ratio in the anagen (active growth phase) and telogen phases (resting phase) was termed the A/T ratio. All data have been reported as mean ± SD and analyzed by one-way analysis of variance (ANOVA). Least Significant Difference (LSD) followed this as the post hoc test, and *p* < 0.05 was considered the significance level.

### 4.8. LC-MS/MS Analysis

This stage was preceded by identifying the active sub-fraction compound resulting from anti-alopecia activity using LC-MS/MS QTOF. The liquid chromatography system on the column was set at 50 °C and 25 °C at room temperature. A total of 5 mM ammonium formic acid in water (eluent A) and 0.05% formic acid in acetonitrile (eluent B) were used as solvents with a flow rate of 0.2 mL/min (step gradient), and the sample was filtered through a 0.2 µm filter, and the injection volume was 5 µL. Mass spectrometry quadrupole time-of-flight equipped with an electrospray ionization (ESI) system in positive ion mode *m*/*z* 50–1200, with optimization parameters; source temperature 100 °C, cone gas flow 0 L/h, collision energy 4 V (low energy), ramp collision energy 25–60 V (high energy), desolvation gas flow, and the temperature was set at 793 L/h and 350 °C. The graph of the analysis results was processed with Masslynx software version 4.1 and traced with a built-in library database of instruments (Waters Corp. Milford, MA, USA).

### 4.9. Anti-Alopecia Activity: In Silico Studies (Molecular Docking Simulation)

The structures of the active sub-fraction compounds were retrieved from the ChemSpider compound database (https://chemspider.com (accessed on 22 August 2022)) and downloaded in a mol format. The structures were sketched using Marvin Sketch 22.14 (ChemAxon, Budapest). Additionally, the compound ligands were prepared using the AutoDock Tools program and stored in PDB format. The selected target receptor was androgen (PDB ID: 4K7A), whose protein crystal structure was obtained from http://www.rscb.org/pdb/ (accessed on 28 August 2022). This receptor predicted the activity of an androgen [53,54]. Meanwhile, the docking process used standard docking protocols, AutodockTools 1.5.6 [57,58,59]. The method was validated by removing natural ligands (DHT and Minoxidil) at the androgen 4K7A receptor. This was conducted by adjusting the grid box position (40 × 40 × 40), coordinates of −2.592, 0.864, and −6.729 (as the *x*, *y*, and *z* center of mass, respectively), redocking it, and producing an RMSD value of 1.732 Å. These results indicate that the software docking method is valid, and the results are successful when the RMSD value is ≤3 Å [59]. The docking conformation as the best pose was selected with a low binding energy of less than 0.1 nm in positional root mean square deviation (RMSD). Receptor and ligand interactions and amino acid bonds with ligands were visualized using the Biovia Discovery Studio Visualizer 2021. The pharmacokinetic properties and the prediction of the toxicity of the test compound ligand as new drug candidates should be obtained using the Pre-ADMET at https://preadmet.qsarhub.com/ (accessed on 5 September 2022).

### 4.10. Anti-Alopecia Activity: In Silico Studies (Molecular Dynamics Simulation)

Molecular dynamics simulations were performed on all conformations with the lowest energy in the docking results using the solution builder on CHARMM-GUI [60,61]. The ligands were parameterized using PDB coordinates, and the protein used the CHARMM36 force field [62]. Protein and ligand preparation results were further processed using GROMACS 2019 [63], which consisted of equilibration and minimization. Then, molecular dynamics production was carried out for 100 ns. The interpretation is displayed in the form of a graph of the root mean square deviation (RMSD) on the backbone, the root mean square fluctuation (RMSF) on C-alpha, and the solvent-accessible surface area (SASA) on proteins using QtGrace software.

## 5. Conclusions

A study in vivo acquired the results of a statistical histology test that FEA-3 25% sub- fractions of Noni fruit could potentially act as anti-alopecia with high follicle density and increased anagen phase. Furthermore, the FEA-3 subfraction was identified by LC-MS/MS and found six compounds containing alkaloid skeletons, such as indole, piperazine, piperidine, oxazolidine, and pyrimidine. These results are highly relevant to Minoxidil as a standard, also having an alkaloid skeleton (piperidine–pyrimidine).

Predicated on an in silico test, the compounds that acted as an anti-alopecia using androgen receptor (PDB: 4K7A) were Compounds **2**, **3**, and **4** supported by ADME-Tox properties with good results. Molecular dynamics confirmed these results; the RMSD, RMSF, and SASA values gave good average results, with Compound **2** showing a better stability profile than Compounds **3** and **4**. Hence, it can be concluded that compounds from the FEA-3 sub-fraction of Noni fruit (referring to the in silico prediction, an alkaloid group) can be used as new anti-alopecia agents in treating alopecia. This study reports and reveals, for the first time, a group of alkaloids from the Noni fruit plant that plays an essential role as candidates for anti-alopecia drugs with supporting in vivo and in silico data.

## Figures and Tables

**Figure 1 pharmaceuticals-15-01557-f001:**
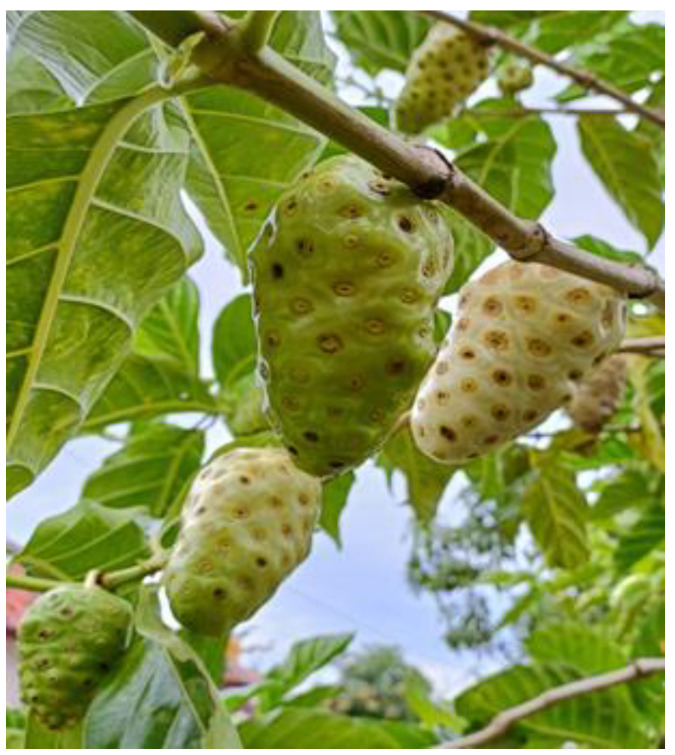
Noni fruit (*Morinda citrifolia* L.).

**Figure 2 pharmaceuticals-15-01557-f002:**
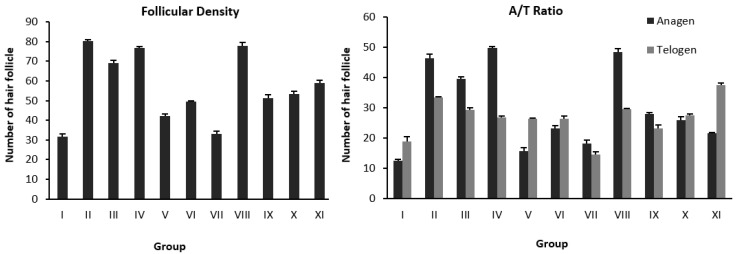
Graph of the calculation of hair follicular density and A/T ratio.

**Figure 3 pharmaceuticals-15-01557-f003:**
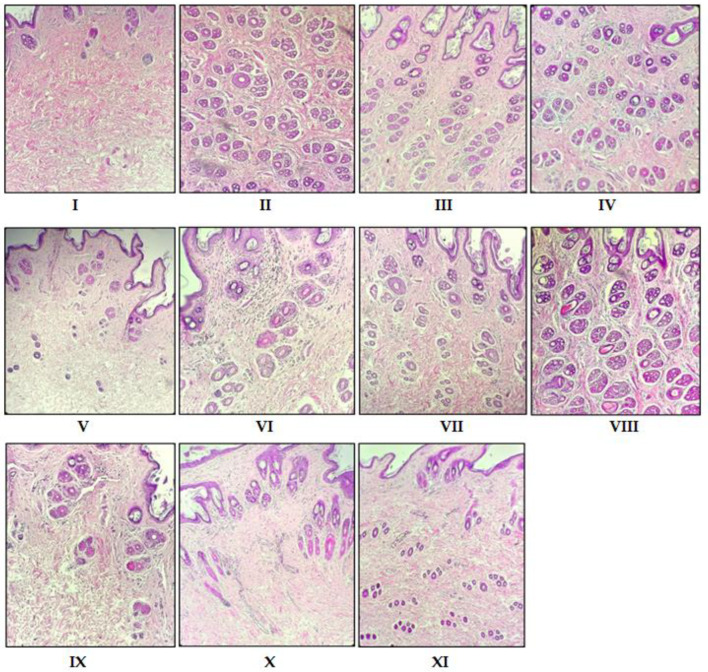
Photomicrograph comparison of hair follicles for each group; (**I**) DHT+vehicle, (**II**) DHT+Minoxidil, (**III**) DHT+FW 25%, (**IV**) DHT+FEA 25%, (**V**) DHT+FH 25%, (**VI**) DHT+FEA-1 25%, (**VII**) DHT+FEA-2 25%, (**VIII**) DHT+FEA-3 25%, (**IX**) DHT+FEA-4 25%, (**X**) DHT+FEA-5 25%, (**XI**) DHT+FEA-6 25% (magnification 20×).

**Figure 4 pharmaceuticals-15-01557-f004:**
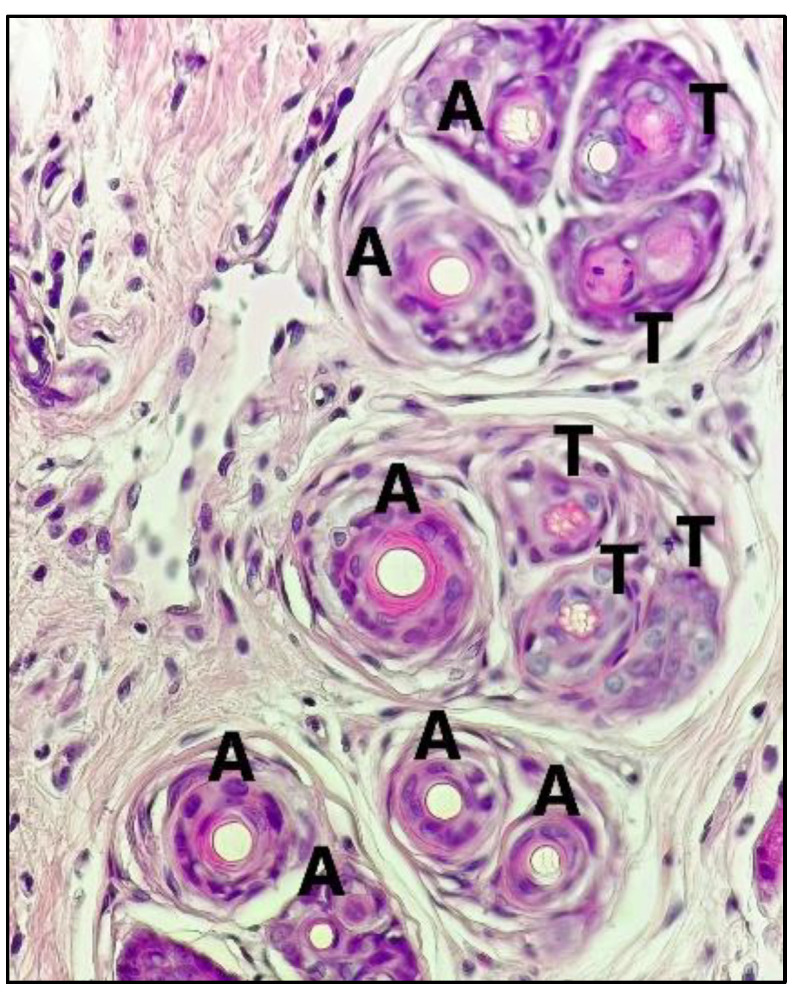
Photomicrograph profile of anagen (A) and telogen (T) (magnification 40×).

**Figure 5 pharmaceuticals-15-01557-f005:**
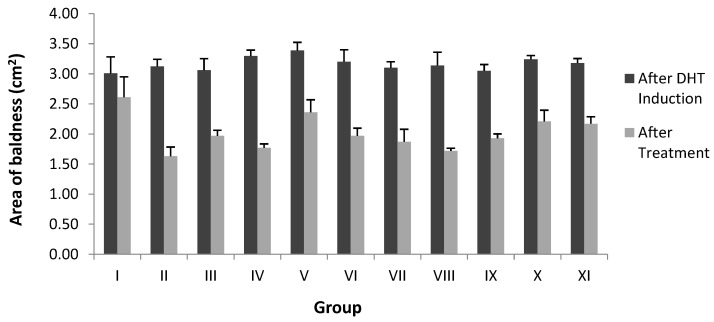
Graph of comparison area of baldness after DHT induction and after treatment.

**Figure 6 pharmaceuticals-15-01557-f006:**
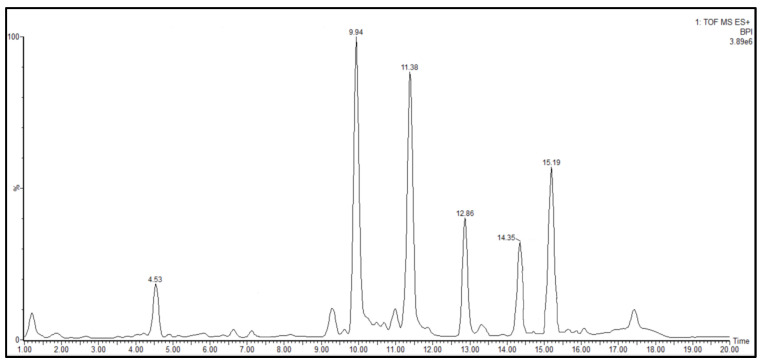
The LC peak chromatogram of FEA-3 sub-fraction.

**Figure 7 pharmaceuticals-15-01557-f007:**
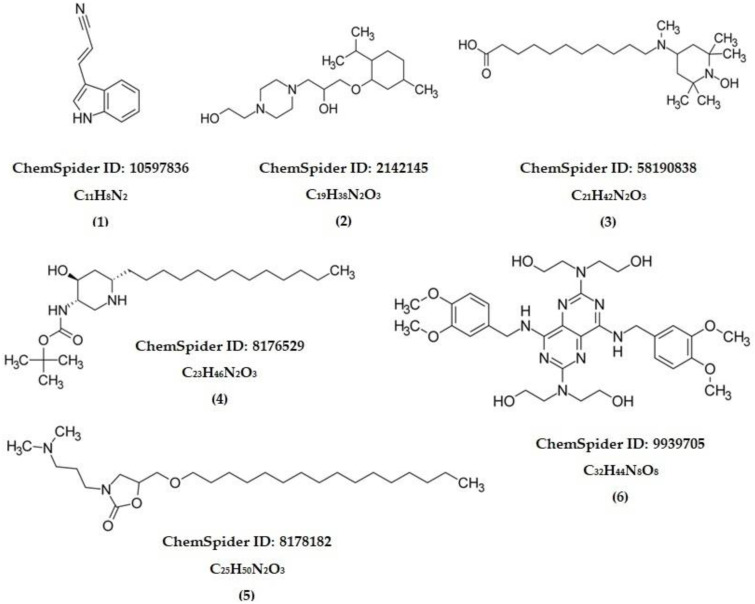
The chemical structures of FEA-3 sub-fraction.

**Figure 8 pharmaceuticals-15-01557-f008:**
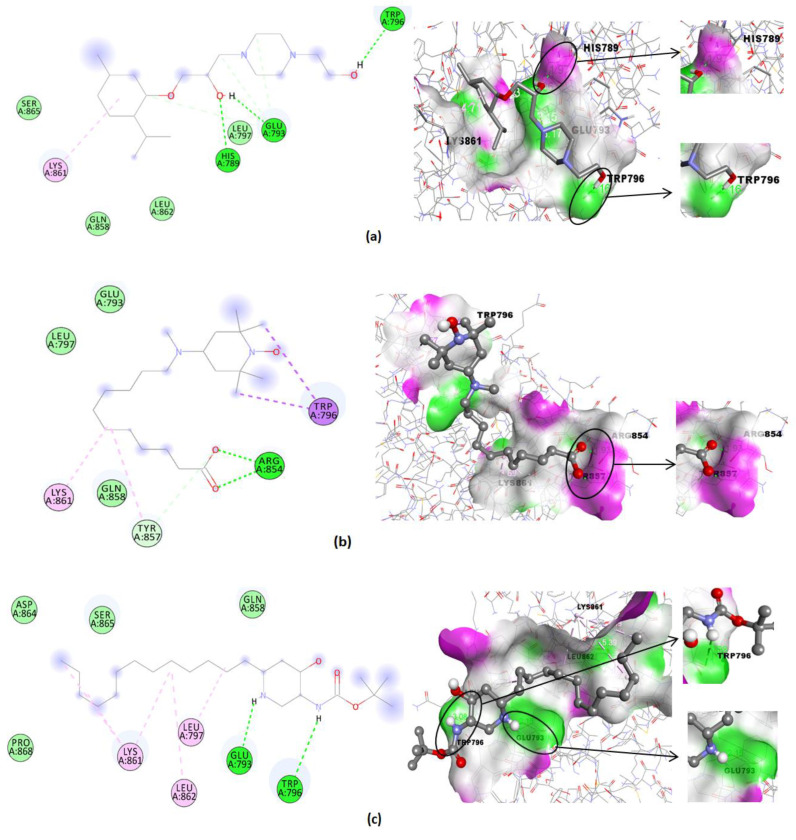
Visualization of amino acid interaction and H-bond surface between the androgen receptor and (**a**) Compound **2**, (**b**) Compound **3**, (**c**) Compound **4**, and (**d**) Minoxidil (standard).

**Figure 9 pharmaceuticals-15-01557-f009:**
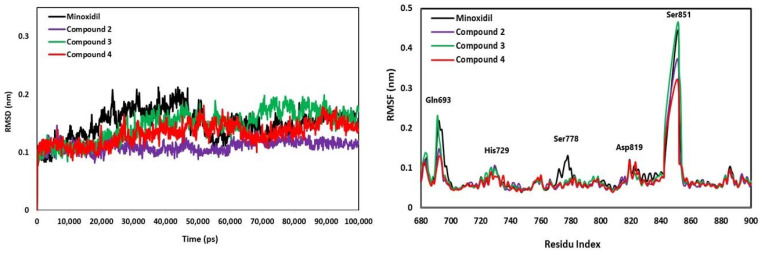
The graph of RMSD and RMSF from each compound in complex with the androgen receptor.

**Figure 10 pharmaceuticals-15-01557-f010:**
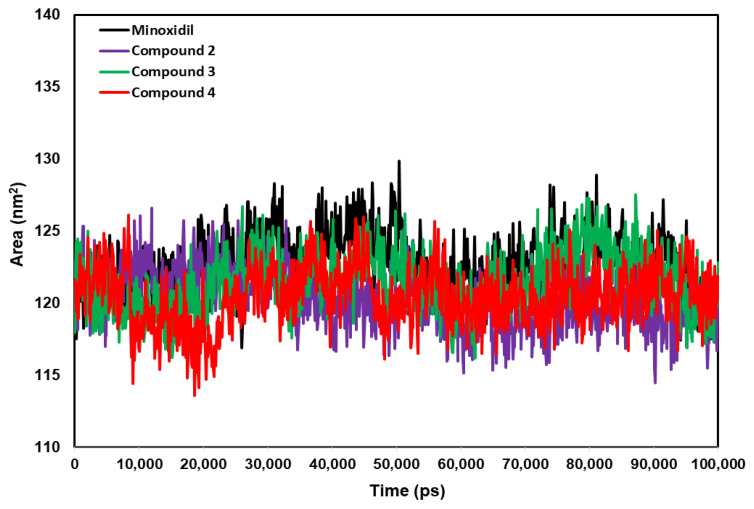
The graph of SASA area for androgen receptor–compound complexes during 100 ns of simulation.

**Table 1 pharmaceuticals-15-01557-t001:** The results of standardization of Noni fruit extract.

Specific and Non-Specific Parameters	Results	Standard [36]
Shapes	Viscous extract	Viscous extract
Color	Dark brown	Dark brown
Odor	Distinctive odor	Distinctive odor
Taste	Slightly bitter	Bitter
Yield (%)	14.65	>10
Water content (%)	8376	≤10
Total ash content (%)	0.7	≤0.8
Acid insoluble ash content (%)	0.08	≤0.1

**Table 2 pharmaceuticals-15-01557-t002:** Phytochemical screening of FW, FEA, and FH of Noni fruit.

Phytochemical Screening	Test	FW	FEA	FH
Alkaloid	Dragendorf	+	+	-
	Mayer	+	+	-
	Wagner	+	+	-
Flavonoid	Shinoda	+	+	-
Phytosterols	Liebermann–Burchard	-	-	+
Anthraquinones	Brontager	-	-	+
Saponins	Foam	+	+	-
Tannins	Ferric Chloride	+	+	-

FW: fraction of water; FEA: fraction of ethyl acetate; FH: fraction of *n*-hexane (+ = Presence; - = Absence).

**Table 3 pharmaceuticals-15-01557-t003:** Hair follicular density and A/T ratio of skin section after administration of the samples topically.

Stage	Group	Treatment	Hair Follicular Density		Total of Hair Follicular Density	A/T Ratio
			Anagen	Telogen		
Fraction	I	DHT (s.c.)+vehicle (NaCMC 1%)	12.55 ± 0.38	19.00 ± 1.53	31.55 ± 1.64 *	(1/1.50)
	II	DHT (s.c.)+Minoxidil	46.56 ± 1.17	33.56 ± 0.19	80.12 ± 1.02 **	(1.39/1)
	III	DHT (s.c.)+FW 25%	39.67 ± 0.67	29.45 ± 0.69	69.12 ± 1.35 **	(1.35/1)
	IV	DHT (s.c.)+FEA 25%	49.78 ± 0.51	27.00 ± 0.33	76.78 ± 0.84 **	(1.84/1)
	V	DHT (s.c.)+FH 25%	16.78 ± 0.88	25.33 ± 0.33	42.11 ± 1.07 *	(1/1.68)
Sub-fraction	VI	DHT (s.c.)+FEA-1 25%	23.33 ± 1.07	26.33 ± 1.00	49.66 ± 0.33 *	(1.13/1)
	VII	DHT (s.c.)+FEA-2 25%	18.22 ± 1.07	14.67 ± 0.88	32.89 ± 1.71 *	(1.24/1)
	VIII	DHT (s.c.)+FEA-3 25%	48.44 ± 1.17	29.55 ± 0.38	78.00 ± 1.52 **	(1.64/1)
	IX	DHT (s.c.)+FEA-4 25%	28.00 ± 0.58	23.22 ± 1.26	51.22 ± 1.68 *	(1.21/1)
	X	DHT (s.c.)+FEA-5 25%	25.89 ± 1.17	27.56 ± 0.51	53.45 ± 1.39 *	(1/1.06)
	XI	DHT (s.c.)+FEA-6 25%	21.55 ± 0.38	37.55 ± 0.69	59.11 ± 1.07 *	(1/1.74)

The results represented as mean ± SD (*n* = 3); * *p* < 0.05, significance vs vehicle; ** *p* < 0.05, significance vs Minoxidil. (DHT: Dihydrotestosterone; FW: fraction of water; FEA: fraction of ethyl acetate; FH: fraction of *n*-hexane; s.c.: subcutaneous; A/T Ratio: anagen/telogen ratio).

**Table 4 pharmaceuticals-15-01557-t004:** Area of baldness after DHT induction and after treatment of a sample.

Stage	Group	Area of Baldness (cm^2^)	
		After DHT Induction	After Treatment of Sample
Fraction	I	3.01 ± 0.23	2.61 ± 0.34 *
	II	3.12 ± 0.11	1.63 ± 0.15 **
	III	3.06 ± 0.19	1.97 ± 0.09 **
	IV	3.29 ± 0.09	1.77 ± 0.06 **
	V	3.38 ± 0.14	2.36 ± 0.20 **
	Average	3.18 ± 0.16	2.07 ± 0.17
Sub-fraction	VI	3.20 ± 0.11	1.97 ± 0.12 **
	VII	3.04 ± 0.09	1.87 ± 0.21 **
	VIII	3.10 ± 0.06	1.72 ± 0.04 **
	IX	3.05 ± 0.10	1.93 ± 0.07 **
	X	3.23 ± 0.05	2.21 ± 0.18 **
	XI	3.18 ± 0.07	2.17 ± 0.11 **
	Average	3.15 ± 0.12	1.97 ± 0.12

The results represented as mean ± SD (n = 3), * *p* < 0.05, significance vs vehicle; ** *p* < 0.05, significance vs. Minoxidil.

**Table 5 pharmaceuticals-15-01557-t005:** The LC-MS/MS profile of FEA-3 sub-fraction.

Compound	RT (Min)	Formula	Observed *m*/*z*	Abundance
**1**	4.53	C_11_H_8_N_2_	169.0765	+
**2**	9.94	C_19_H_38_N_2_O_3_	343.2945	+++
**3**	11.38	C_21_H_42_N_2_O_3_	371.3281	+++
**4**	12.86	C_23_H_46_N_2_O_3_	399.3608	++
**5**	14.35	C_25_H_50_N_2_O_3_	427.3687	++
**6**	15.19	C_32_H_44_N_8_O_8_	669.3326	++

+++: high abundance, ++: abundant, +: low abundance (RT; Retention Time; *m*/*z*: mass-to-charge ratio).

**Table 6 pharmaceuticals-15-01557-t006:** The alkaloid compounds of FEA-3 with LC–MS/MS.

Compound	Molecule	Alkaloid Skeleton
**1**	(E)-3-(1H-Indol-3-yl)acrylonitrile	Indole
**2**	1-[4-(2-Hydroxyethyl)-1-piperazinyl]-3-[(2-isopropyl-5-methylcyclohexyl)oxy]-2-propanol	Piperazine
**3**	11-[(1-Hydroxy-2,2,6,6-tetramethyl-4-piperidinyl)(methyl)amino]undecanoic acid	Piperidine
**4**	2-Methyl-2-propanyl [(3S,4S,6S)-4-hydroxy-6-tridecyl-3-piperidinyl]carbamate	Piperidine
**5**	3-[3-(Dimethylamino)propyl]-5-[(hexadecyloxy)methyl]-1,3-oxazolidin-2-one	Oxazolidine
**6**	2,2′,2″,2‴-({4,8-Bis[(3,4-dimethoxybenzyl)amino]pyrimido[5,4-d]pyrimidine-2,6-diyl}dinitrilo)tetraethanol	Pyrimidine
Standard	3-hydroxy-2-imino-6-piperidin-1-ylpyrimidin-4-amine (Minoxidil)	Piperidine–pyrimidine

**Table 7 pharmaceuticals-15-01557-t007:** Molecular docking simulation results.

Compound	Binding Affinity (Kcal/mol)	Hydrogen Bonds	Hydrogen Bonds Distance (Å)	Nearest Amino Acid Residues
**1**	−3.45	-	-	Glu793-Gln858-Tyr857-Lys861-Leu862-Leu797
**2**	−4.99	His789-Glu793-Trp796	1.97; 1.86; 2.16	Ser865-Gln858-Leu862-Leu797-Lys861
**3**	−4.60	Arg854(2)	1.93; 1.74	Gln858-Lys861-Tyr857-Glu793-Leu797-Trp796
**4**	−4.57	Glu793-Trp796	2.15; 3.08	Asp864-Ser865-Pro868-Gln858-Lys861-Leu797-Leu862
**5**	−2.76	His789-Lys861	2.94; 2.27	Glu793-Gln858-Ser865-Asp864
**6**	−3.39	Leu862-Gln858	3.07; 2.27	Lys861-Glu793-Leu797-Tyr857-Arg854-Trp796
Minoxidil (standard)	−4.83	His789	2.69	Gln858-Glu797-Tyr857-Leu862-Trp796-Lys861-Leu797

**Table 8 pharmaceuticals-15-01557-t008:** Results of ADME-Tox properties.

Compound	Absorption		Distribution		Mutagenic	Carcinogenic
	HIA (%)	SP (Log K_P_)	BBB (Log BB)	PPB (%)		
**2**	91.84	−2.77	1.40	43.48	yes	no
**3**	95.62	−1.255	1.53	93.21	no	no
**4**	89.07	−0.851	6.87	95.13	no	no
Minoxidil	82.11	−3.930	0.26	58.38	yes	no

(HIA: human intestinal absorption; SP: skin permeability; BBB: blood–brain barrier; PPB: plasma protein binding).

## Data Availability

Data is contained within the article and Appendix A.

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
