# Peer review of "Anti-Alopecia Activity of Alkaloids Group from Noni Fruit against Dihydrotestosterone-Induced Male Rabbits and Its Molecular Mechanism: In Vivo and In Silico Studies"

_pharmaceuticals, 2022, doi:10.3390/ph15121557_

Round 1

Reviewer 1 Report

The authors performed a nice study regarding using Noni fruit for treatment of alopecia. However, the author needs to address the following comments;

1.   Extensive editing of English language and style are required.

2.   The introduction part should be improved:

·         The introduction doesn’t provide sufficient background.

·         More data about Noni fruit should be added.  

·         Please add an image of Noni fruit.

·         The previous studies that discussed using other methods and drugs in the treatment of alopecia should be involved in more details.

·         Please clarify the advantages of using Noni fruit.

3.   The writing of methodology and results sections should be improved.

4.   The data and specifications of the used equipment should be added.

5.   The abbreviations should be added below the tables and figures.

6.   The units of different measurements should be added within the paragraphs and tables.

7.   ANOVA is required in the results section to investigate the occurrence of significant difference between the studied groups.

8.   The values of standard deviation should be added to the reported data. Some data are written without SD.

9.   Figure 6 should be modified.

Reviewer 2 Report

Study is well performed to define the anti-alopecia effect of Noni  fruit compounds but requires attention on following points

Comments

1)      Abstract: FEA abbreviation should be defined clearly when used first. It should be strictly followed throughout the manuscript.

2)      Page 2, 5th paragraph “DHT was inducer to produce an alopecia animal model” reframe appropriately. There are many such errors in the manuscript and requires attention.

3)      Authors should clarify in the manuscript, whether the hairfollicle density analysis includes both anagen and telogen follicles. If yes, give separate figures for both in the Table 3. The manuscript  should also define the histopathological features of anagen and telogen.  

4)       Since DHT induced animal models of alopecia showed widespread baldness, have authors ensured that that site matched skin were used for histopathological analysis?  Is there any site specific difference observed in the distribution of anagen and telogen hairs. Authors should share the representative photograph of rabbits showing extent of alopecia. The Photomicrograph shown in figure 2, gives an impression that the sections were not site matched and section belongs to different sites?

5)      It will be interesting to see the effect on hair follicles in the longitudinal skin section and staining for follicular stem cell marker(s).    

6)      Is there any specific reason to use rabbits and not mouse model?
